# Modeling and Predicting Passenger Load Factor in Air Transportation: A Deep Assessment Methodology with Fractional Calculus Approach Utilizing Reservation Data

**Kevser Şimşek [1,*](ID), Nisa Özge Önal Tuğrul [1](ID), Kamil Karaçuha [2](ID), Vasil Tabatadze [1](ID) and Ertuğrul Karaçuha [1](ID)**

[1] Informatics Institute, Istanbul Technical University, Istanbul 34467, Turkey; onal16@itu.edu.tr (N.Ö.Ö.T.); tabatadze@itu.edu.tr (V.T.); karacuhae@itu.edu.tr (E.K.)
[2] Faculty of Electrical and Electronics Engineering, Istanbul Technical University, Istanbul 34467, Turkey; karacuha17@itu.edu.tr
[*] Correspondence: simsek18@itu.edu.tr; Tel.: +90-212-463-0000

**Abstract:** This study addresses the challenge of predicting the passenger load factor (PLF) in air transportation to optimize capacity management and revenue maximization. Leveraging historical reservation data from 19 Turkish Airlines market routes and sample flights, we propose a novel approach combining deep assessment methodology (DAM) with fractional calculus theory. By modeling the relationship between PLF and the number of days remaining until a flight, our method yields minimal errors compared to traditional techniques. Through a continuous curve constructed using the least-squares approach, we enable the anticipation of future flight values. Our analysis demonstrates that the DAM model with a first-order derivative outperforms linear techniques and the Fractional Model-3 in both modeling capabilities and prediction accuracy. The proposed approach offers a data-driven solution for efficiently managing air transport capacity, with implications for revenue optimization. Specifically, our modeling findings indicate that the DAM wd model improves prediction accuracy by approximately 0.67 times compared to the DAM model, surpassing the fractional model and regression analysis. For the DAM wd modeling method, the lowest average mean absolute percentage error (AMAPE) value achieved is 0.571, showcasing its effectiveness in forecasting flight outcomes.

**Keywords:** LF prediction; fractional analysis in engineering problems; deep assessment methodology; linear regression analysis; fractional calculus; air transportation





## 1. Introduction

The passenger load factor (PLF) is a metric that refers to the percentage of seats that are occupied by passengers. It also determines how well an airline's demand and capacity management operations are working, so it is commonly used to assess efficiency and effectiveness.

Although there have been several studies on the airline business, only a few have attempted to estimate the load factor parameter that is commonly used to describe an airline's capacity management performance [1]. The load factor for the American airline industry was calculated using the number of travel agent locations using each computerized reservation system, the average length in miles of all airline flights between city pairs, the number of departures for each carrier, advertising expenses, and the change in vehicle miles. The number of departures for each carrier, as well as the cost of advertising, were found to be significant factors in explaining the load factor for American Airlines [2]. Pegels and Yang tested a domestic air transportation model in the United States. In this model, the load factor is a dependent variable that is determined by available seat miles, total assets, and advertising costs [3]. In another paper, two gravity models are presented for estimating air passenger volume between city pairs, which can be applied to city pairs where no air

service is established, where historical data is not available, or for which factors describing the current service level of air transportation are not accessible or accurately predictable, and both models show a good fit to the observed data and are statistically tested and validated [4].

Stochastic models have been established to determine load factors, which is the best-fit trend for Europe's North Atlantic and mid-Atlantic flights in the Association of European Airlines [5]. The load factor has both periodic and serial correlations. For serial and periodic correlations, two distinct models were developed. The Prais–Winsten technique for serial correlations and dynamic temporal effects for periodic correlations were later integrated into the models [6].

In the study of airlines, there are lots of restrictions that can be considered while explaining load factors. The inherent uncertainty associated with predicting these restrictions is what makes this challenge so difficult, leaving human decision-makers dependent on expertise to develop efficient air traffic capacity management measures. Some of the variables are the reservation system, the average length of miles between city pairs, and the number of departures for each carrier. [7]. The price and demand balance is taken into account with the gradual sale of the tickets to maintain a high load factor rate, as well as the integrity of network connections [8]. Based on the first investigation, it was discovered that passenger load varies significantly depending on parameters such as airline type (full-service vs. low-cost), aircraft type, destination, etc. [9].

In airline operations, knowing the future values of the load factor that has been agreed upon to identify the line is critical. This line is determined by several things. One of them is that aviation operations are controlled in a loop, and the load factor in the aviation industry is not stable [10]. Another is that airlines use dynamic capacity management to meet and absorb demand, and the airlines can alter the type of aircraft they use at any time [11]. The consumer is strong due to the fierce competition between end-market free-market airlines and the online reservation system (ORS). As a result, dynamic decisions are made based on airway efficiency, which affects the load factor [12–14].

As mentioned above, many parameters and issues need to be taken into account during modeling, analyzing, or monitoring the current status of the airlines and the prediction of forthcoming days. In the present study, fractional calculus is employed for modeling since such a method provides more flexible mathematical modeling. Fractional calculus, which is more broadly defined as the calculus of integrals and derivatives of any arbitrary real or complex order, arose from a query posed to Wilhelm Leibniz (1646–1716) by French mathematician Marquis de L'Hopital (1661–1704) in 1695 [15–18]. Marquis de L'Hopital asked what would happen if the derivative order becomes 0.5. "This is an apparent paradox from which, one day, useful consequences will be drawn. . ." was Leibniz's response to the question [19].

Over the last 50 years, many people with different professions have shown that fractional derivatives and integrals contain crucial information about the systems they are investigating. The fractional derivative, in particular, gives useful information for complex problems, processes with memory, and heredity. In control theory, mechanic, economic, finance, and electromagnetic fractional calculations are frequently utilized [15–25]. AlBaidani's study, for instance, analyzes the time-fractional Kawahara and modified Kawahara equations, which are crucial for modeling nonlinear water waves and signal transmission. There are two new techniques offered: the homotopy perturbation transform and the Elzaki transform decomposition. To handle nonlinear terms efficiently, fractional derivatives in the Caputo sense are combined with Adomian and He's polynomials [26]. In a further study, modeling based on fractional-order derivatives is utilized to predict future trends in confirmed cases and deaths from the COVID-19 outbreak in India through October 2020. A mathematical model based on a fractal fractional operator is built to explore the dynamics of the disease epidemic, accounting for various transmission channels and the role of environmental reservoirs [27]. Another study explains how to create mathematical models for claydate constructions that consider the material's fractal structure to solve the

heat conduction problem. The use of fractional order integro-differentiation machinery explains its fractal structure [28].

A review article focuses on variable-order fractional differential equations (VO-FDEs), which are utilized to explore dynamics dependent on time, space, or other variables. The study aims to survey the recent literature on definitions, models, numerical methods, and applications of VO-FDEs [29]. Podlubny's book provides thorough coverage of essential topics such as special functions, the theory of fractional differentiation, analytical and numerical methods for solving fractional differential equations, and practical applications. This is useful for those seeking more in-depth information about fractional derivatives and fractional-order mathematical models. It serves as a resource for both pure mathematicians and applied scientists, offering inspiration for additional study and the rapid application of concepts from fractional calculus [30]. We developed a mathematical approach in this work that uses fractional calculus theory to analyze and explore the components that are continuously related to the modeling and forecasting of the passenger load factor. Using information from two years of reservations, group sales data, calendar information, weekly dates, trend difference between the current year and the previous year, past load factor information, and load factor information from the same period of the previous year, we developed the linear model. In the fractional Model-3, DAM, and DAM wd models using the reservation rate data of the year 2016, by the least-squares method, we developed a continuous curve valid for any time interval. The theory, numerical results of the proposed theory, and comparisons with various modeling approaches including machine learning (linear) are discussed in this work.

In forecasting, two main stochastic numerical methods are highly useful: the first one is time series, and the second one is regression analysis. Nominal variables may have varying values inside each time unit in a time series. Factors are another name for nominal variables—for instance, a location, a country, a profession, a province, per capita income, unemployment rates in various nations, job-based income, and so on. The factor effect cannot be accurately measured using a time-series analysis. On the other hand, regression analysis does not account for the dynamic effect of external variables on internal variables. Panel data and the least-squares method appear to be a good combination [25].

PLF is calculated by dividing the revenue passenger kilometers (RPK) by the available seat kilometers (ASK). Hence, assuming the capacity of an airline remains the same, an increase in RPK is directly proportional to an increase in PLF. Its formula is shown in Equation (1) below:

$$PLF = \text{Revenue Passenger Kilometers}/\text{Available Seat Kilometers} \tag{1}$$

The set of objectives of this study can be itemized as follows:
After the flights are opened for sale, predicted PLF on boarding times;

- Understanding the seasonality,
- Revealing the special day's effects,
- Revealing the previous flight's behaviors,
- Board-off country and area-based predictions.

This paper is the first one to adapt the fractional Model-3, DAM, and DAM wd model to air PLF and on the comparison of these models. In our research, we determine the finest fractional order value of the derivative for each factor, which will allow us to develop effective modeling. We also build a sample application using air PLF market area and flight data to put the mathematical models into practice.

The formulation for the modeling and prediction will be provided in the next section. Applications will be provided in the third section. The results and figures to show the comparison of the proposed models are provided in the fourth part.

## 2. The Proposed Approaches

### 2.1. The Modeling and Predicting with Machine Learning—Linear Regression Model

2.1.1. Panel Data Benefits

This study aimed to forecast the passenger load factor (PLF) by using information on two-year reservations, group sales data, calendar information, weekly dates, trend differences between the current and previous year, and load factor information of the same period of the previous year. Panel data allow individual heterogeneity control. Panel data suggest that every flight, flight destination, flight departure, and arrival are heterogeneous [31]. Time-series and cross-section studies that fail to account for this heterogeneity risk producing skewed results [32].

More information, more variety, less collinearity across variables, more degrees of freedom, and more efficiency are all advantages of panel data [33]. The remaining days of departure within the reservation sales data, for example, have a significant degree of collinearity in airline data. Panel data are preferable for studying adjustment dynamics. Even though cross-sectional distributions are steady, they include a lot of variation. It allows us to figure out who benefits from development. It also allows us to determine whether poverty and deprivation are transient or long-term, a subject known as income dynamics. In reality, panels can link an individual's past experiences and conduct to future events and behavior [34].

2.1.2. Data Preparation and Metadata Creation

The data processing procedure is the first phase, which involves selecting specific reservation days from the time-series format to utilize in the panel format. Otherwise, 1.8 billion rows of data for all reservation days would have to be processed, and it will take days with a single server that does not distribute models in common data frame-like formats at this moment. Using the group sales data frame, the lag of load factor information, and the calendar data frame, the final metadata for analysis and modeling will be created in this phase. Tables 1–3 below display data frames:

**Table 1.** An Example of Reservation Lag Data Frame.

| Origin_YMD | Flight_Number | Board_Point | Off_Point | Compartment | Remaining_Day | Seat_Sold |
|---|---|---|---|---|---|---|
| 20160101 | 2328 | IST | ADB | Y | 0 | 0.63 |
| 20160101 | 2328 | IST | ADB | Y | 1 | 0.57 |

**Table 2.** An Example of Group Sales Data Frame.

| Origin_YMD | Flight_Number | Board_Point | Off_Point | Compartment | Remaining_Day | Group_Seat_Sold |
|---|---|---|---|---|---|---|
| 20160101 | 2328 | IST | ADB | Y | 0 | 0.43 |
| 20160101 | 2328 | IST | ADB | Y | 1 | 0.37 |

**Table 3.** An Example of Lag of Load Factor Data Frame.

| Origin_YMD | Flight_Number | Board_Point | Off_Point | Compartment | LF |
|---|---|---|---|---|---|
| 20160101 | 2328 | IST | ADB | Y | 0.53 |
| 20160103 | 2328 | IST | ADB | Y | 0.46 |

From the initial day of booking until the last day, this data frame is a daily time data frame format.

This data frame has an upward effect on flights over time and is a daily time data frame format from the first day of booking to the last day.

The reservation lag data frame serves as the source for Table 3. When a plane is first launched, booking information is quite helpful, and the typical departure times of that

same jet are also significant. For example, if we look at two distinct flights that have 30 days to go, we can see that 85 percent of these flights are mostly located in the province, with the remaining 65 percent or so are outside of it. It is anticipated that the model we created will yield distinct outcomes for these two flights. The latency of load factor information is the data that will provide us with this information.

### 2.1.3. Panel Data Creation

The panel data structure can be created once the metadata has been created. An item and time variables must be generated in the panel data structure shown in Table 4 as an example.

**Table 4.** An Example of Reservation Table.

| Item | Time | Y | X1 | X2 | X3 |
|------|------|-----|-----|-----|-----|
| A | 20160101 | 0.78 | 0.67 | 0.35 | 0.63 |
| B | 20160102 | 0.87 | 0.37 | 0.29 | 0.57 |

The panel data frame in our case looks like Table 5. We start the data preparation process with the reservation lag table. First, we select specific reservation days to use in the panel format from the time-series format. After generating the lag of load factor data frame, we combine it with the group sales data frame and the calendar data frame. The board-country and off-country variables will be integrated as an item variant, and the model will be changed at a higher level than the airport level. This is being undertaken to improve the model's functioning performance. Table 5 displays the table format. Table 5 labels are as follows: Id refers to the flight route, Origin YMD is the departure day, Board Point refers to the departure point, Off Point refers to the arrival point, Compartment includes business and economy classes, LF is load factor, and RES1, RES2, ... mean the reservation rates of the flight. For instance, RES3 means the reservation rate before 3 days to boarding time.

**Table 5.** The Panel Data Matrix of Our Study.

| Id | Origin YMD | Flight Number | Board Point | Off Point | Compartment | LF | RES1 | RES2 | RES3 |
|------|------|------|------|------|------|------|------|------|------|
| Y-TRTR | 20160101 | 2328 | IST | ADB | Y | 0.844 | 0.701 | 0.735 | 0.776 |
| Y-TRTR | 20160102 | 2328 | IST | ADB | Y | 0.882 | 0.819 | 0.819 | 0.844 |

As shown in Table 5, the key variable is our item variable and Origin YMD is our time variable.

### 2.1.4. Linear Regression Modeling

After detailed data analysis, we are in the phase of modeling. The R programming language is used during the modeling process and the development of the images.

The formula for calculating the load factor is shown in Equation (2). The target variable is established when this calculation is completed for each flight and cabin class. When we meet multi-leg flights, we handle the ratio problem by weighing kilometers with this method.

$$Load\ Factor = \sum_{i=1}^{t} \left( \frac{(Number\ of\ Carried\ Passenger)_t * Distance_t}{(Available\ Seat)_t * Distance_t} \right) \tag{2}$$

The entire year 2015 will be used as training data throughout the modeling phase. In 2016, we will put our model to the test. Forty percent of the training data will be used as test data, while the other part will be utilized as training data [35]. The mathematical formula we use to calculate the load factor target variable is as follows:

$$LF_{it} = \beta_0 + \beta_1 GLF_{it} + \beta_2 WCLF_{it} + \beta_3 LLF_{it} + \beta_4 BCOCLF_{it} + \sum_{k}^{340} RES_{it}\beta_r + \sum_{m}^{12} M_{it}\beta_m + \sum_{y}^{4} YR_{it}\beta_y$$
$$+\sum_{d}^{7} DOW_{it}\beta_d$$

(3)

where

- $k$ = 1, 2, 3, 4, 7, 10, 15, 20, 30, 45, 60, 90, 120, 340 (RES parameter),
- $m$ = 1, 2, 3,..., 12 (month),
- $y$ = 1, 2, 3, 4 (year),
- $d$ = 1, 2, 3,..., 7 (day of week),
- $\beta_0$ = Y-intercept (base load percentage),
- $\beta_1$ = group LF coefficient,
- $\beta_2$ = LF wait count coefficient,
- $\beta_3$ = 365 days lag of LF coefficient,
- $\beta_4$ = board-country–off-country LF coefficient,
- $\beta_r$ = RES $i$ coefficient,
- $\beta_y$ = year $i$ coefficient,
- $\beta_m$ = month $i$ coefficient,
- $\beta_d$ = day of week coefficient,
- $GLF$: group LF,
- $WCLF$: wait count LF,
- $LLF$: 365 days lag of LF,
- $BCOCLF$: board-country–off-country LF,
- $RES$: reservation,
- $M$: month,
- $YR$: year,
- $DOW$: day of week.

By including different $\beta$ values according to added parameters like day of week, group LF coefficient, etc. that will affect the load factor, we try to model and predict the number of carried passenger values more accurately than the usual formula given in Equation (2). There are distinct departure days for each flight while forecasting the coming days in the modeling. The following function was used to dynamically program in this case. With this function, we can open up what we wish to achieve: for example, if the flight is scheduled to depart in 10 days, RES15 will be used to calculate the model. In addition, for example, if the plane is scheduled to depart in 25 days, the RES30 will be included in the model. As a result, multiple models will be constructed depending on the RES values. RES0, RES1, RES2, RES3, RES4, RES5, RES6, RES7, RES8, RES9, RES10, RES12, RES15, RES20, RES30, RES45, RES60, RES90, RES120, RES200, and RES340 are the 21 different model outcomes.

### 2.2. Modeling with Fractional Calculus

For the fractional calculus method used in the present study, the information of the 2015 daily reservation data that includes 19 marketing area groups and the number of days to the flight time is used.

In the study, the fractional model named Fractional Model-3 was used for comparing different $N$ exponent values up to 5, 9, and 15 ($N$: 5, 9, 15) in Equation (5). We obtained results from linear regression, deep assessment, deep assessment with derivative, and fractional Model-3, respectively. The models were compared using the mean absolute percentage error (MAPE).

By using the fractional calculus method, we found a solution to predict all RES values by using the formulation explained in the following sub-sections.

### 2.2.1. Fractional Model

The main motivation is to model the given discrete reservation dataset and obtain a continuous curve representing the dataset with the minimum error. To achieve this goal, first, the fractional derivative technique named Fractional Model-3 is proposed to make use of the fractional calculus' flexibility and hereditary features [36]. The derivative order adds another parameter to the equation, making it easier to optimize the model's results.

In the first step, when examining the Caputo equation, you can see that the fractional-order derivative $\mathfrak{D}_x^\alpha$ is defined in this formula:

$$\mathfrak{D}_x^\alpha g(x) = \frac{d^\alpha g(x)}{dx^\alpha} = \frac{1}{\Gamma(n-\alpha)} \int_0^x \frac{g^{(n)}(k)dk}{(x-k)^{\alpha-n+1}}, \ (n-1 < \alpha < n) \tag{4}$$

where derivative order $\alpha$ and $\alpha \epsilon (0,1)$. $\Gamma(1-\alpha)$ is a Gamma function, and this function is described as follows:

$$\Gamma(1-\alpha) = \int_0^\infty t^{-\alpha} e^{-t} dt.$$

After establishing the derivative, Equation (5) is used to represent a continuous function $f(x)$ that describes the relevant data, such as Turkish Airlines' reservation rates over time.

$$\mathfrak{D}_x^\alpha f(x) = \frac{d^\alpha f(x)}{dx^\alpha} = \sum_{n=1}^\infty a_n \left( n\alpha \right) x^{n\alpha-1} \tag{5}$$

It is worth mentioning that the assumption is based on the Taylor expansion and a continuous function's first derivative. The time is expressed by the letter $x$ in this case. The Laplace transform is used to solve Equation (5). The differential equation is converted to an algebraic equation by executing the transform. Then, by inversion operation, the required function $f(x)$ is obtained as given in Equation (8).

The Laplace transform of Equation (4) is given as below:

$$\mathcal{L}\{f(x)\} = F(s) = \frac{f(0)}{s} + \sum_{n=1}^\infty \frac{a_n(n\alpha)}{s^{\alpha n}} \Gamma(n\alpha) \tag{6}$$

$\mathcal{L}$ and $\mathcal{L}^{-1}$ stand for the Laplace transform and inverse Laplace transform of Equation (7):

$$\mathcal{L}^{-1}\{F(s)\} = f(x) = f(0) + \sum_{n=1}^\infty \frac{a_n \Gamma(n\alpha+1) x^{\alpha(n+1)-1}}{\Gamma(\alpha(n+1))} \tag{7}$$

The summation in Equation (7) must be trimmed to $N$ to compute the function numerically. Equation (8) contains an improved version of Equation (7) [37].

$$f(x) \cong f(0) + \sum_{n=1}^N \frac{a_n \Gamma(n\alpha+1) x^{\alpha(n+1)-1}}{\Gamma(\alpha(n+1))} \tag{8}$$

At this point, theoretically, $(x)$ function is achieved. To obtain the unknowns $(0)$, $a_n$, and $\alpha$ the discrete dataset is employed. The least-squares method is used to find these coefficients. The real data values are designated as follows to perform regression:

$$P_i = [p_0 \ p_1 \dots \ p_K]$$
$$x_i = [x_0 \ x_1 \dots \ x_K]$$

where $i = 0, 1, 2, \dots, K$.

Here, $P_i$ corresponds to the given known value at $x_i$ time. It is clear that $f(x)$, which is defined via the fractional derivative, is considered a good candidate for the problem at hand because of its ability to tolerate data changes by using optimum $\alpha$ values that are

calculated by Matlab script, and thus, it has more accurate modeling results and improves the results more than the traditional choice of $\alpha = 1$.

2.2.2. Deep Assessment Method Formulation

A different methodology is used here, as opposed to in the prior section. The fractional model approach for modeling employs fractional calculus, which enables heredity property by the nature of fractional calculus. Although each model employs fractional calculus and a similar approach to obtaining the differential equations, the deep assessment approach regards the required function as the summation of its previous values and its derivatives with unknown coefficients. In other words, the required function for modeling the data is expressed as polynomials with unknown coefficients in the fractional model. However, in the DAM, the required function is expressed in terms of the series sum of the previous values of the function itself and its derivative with unknown coefficients.

In the DAM wd (deep assessment methodology with derivative) model, a continuous function $g(x)$ is assumed to be the sum of its past values and their first derivatives in this scenario. Such datasets containing reservation information are especially ideal to model in that form from an engineering point of view because reservation rates are significantly tied to their past values and changes.

$$g(x) \cong \sum_{k=1}^{l} \alpha_k g(x-k) + \sum_{k=1}^{l} \beta_k g'(x-k) \tag{9}$$

The first derivative of $g(x-k)$ concerning $x$ is denoted by $g'$. The function $g(x)$ can be expanded as the summation of polynomials with unknown constant coefficients, $a_n$, after assuming Equations (9) and (10). $g(x)$ is considered to be a continuous, differentiable function in this case.

$$g(x) = \sum_{n=0}^{\infty} a_n x^n$$

Then, $g(x-k)$ becomes $\quad g(x-k) = \sum_{n=0}^{\infty} a_n (x-k)^n \tag{10}$

Note that, above, the Taylor expansion is again utilized. The final form of $g(x)$ is given as Equation (11).

$$g(x) \cong \sum_{k=1}^{l} \alpha_k \sum_{n=0}^{\infty} a_n (x-k)^n + \sum_{k=1}^{l} \beta_k \sum_{n=0}^{\infty} a_n n (x-k)^{n-1} \tag{11}$$

After combining $\alpha_k a_n$ as $a_{kn}$ and $\beta_k a_n$ as $b_{kn}$ and approximating Equation (11), Equation (12) is obtained. Here, the truncation from $\infty$ to $M$ is performed. The first derivative of $g(x)$ is obtained after truncation and is given in Equation (12).

$$g(x) \cong \sum_{k=1}^{l} \sum_{n=0}^{M} a_{kn} (x-k)^n + \sum_{k=1}^{l} \sum_{n=0}^{M} b_{kn} n (x-k)^{n-1}$$

$$\frac{dg(x)}{dx} \cong \sum_{k=1}^{l} \sum_{n=1}^{M} a_{kn} n (x-k)^{n-1} + \sum_{k=1}^{l} \sum_{n=1}^{M} b_{kn} n (n-1) (x-k)^{n-2} \tag{12}$$

An arbitrary continuous and differentiable function, $g(x)$, is expressed in terms of polynomials at this point, considering that the function is defined as the sum of its past values and derivatives. To continue, the concept of Caputo's fractional derivative should

be represented by the expression provided in Equation (13). Caputo's fractional derivative formula definition is used throughout the study.

$$\mathfrak{D}_x^\gamma g(x) = \frac{d^\gamma g(x)}{dx^\gamma} = \frac{1}{\Gamma(n-\gamma)} \int_0^x \frac{g^{(n)}(k)dk}{(x-k)^{\gamma-n+1}}, \ (n-1 < \gamma < n) \tag{13}$$

After this, it is best to return to our original goal, which is to model the function $f(x)$ and forecast future values. The assumption is that $f(x)$ models the discrete dataset and satisfies the fractional differential equation below:

$$\frac{d^\gamma f(x)}{dx^\gamma} \cong \sum_{k=1}^l \sum_{n=1}^\infty a_{kn} n(x-k)^{n-1} + \sum_{k=1}^l \sum_{n=1}^\infty b_{kn} n(n-1)(x-k)^{n-2} \tag{14}$$

where $f(x)$ stands for reservation rates, and $x$ corresponds to the time.

There are two goals here. To obtain the unknown constants $a_{kn}$ and $b_{kn}$ provided in the differential equation above, first solve Equation (14). The Laplace transform is used to solve the differential equation. The differential equation is then transformed into an algebraic expression. Then, using inversion, we obtain $f(x)$ as Equation (15):

The differential equation must be solved to find the unknowns. The following is the strategy: first, the Laplace transform must be used, resulting in an algebraic equation rather than a differential equation. To put it another way, the Laplace transform is used to simplify the differential equation to an algebraic equation in Equation (14), and then, by using inverse Laplace transform properties, the final form of $f(x)$ is obtained as

$$f(x,\gamma) \cong f(0) + \sum_{k=1}^l \sum_{n=1}^\infty a_{kn} C_{kn}(x,\gamma) + \sum_{k=1}^l \sum_{n=1}^\infty b_{kn} D_{kn}(x,\gamma) \tag{15}$$

where

$$C_{kn}(x,\gamma) \triangleq \frac{\Gamma(n+1)}{\Gamma(n+\gamma)}(x-k)^{n+\gamma-1}$$
$$D_{kn}(x,\gamma) \triangleq \frac{\Gamma(n+1)}{\Gamma(n+\gamma-1)}(x-k)^{n+\gamma-2}$$

The infinite summing of polynomials is approximated as a finite summation provided to achieve the numerical calculation of Equation (16).

$$f(x,\gamma) \cong f(0) + \sum_{k=1}^l \sum_{n=1}^M a_{kn} C_{kn}(x,\gamma) + \sum_{k=1}^l \sum_{n=1}^M b_{kn} D_{kn}(x,\gamma) \tag{16}$$

Then, the second aim needs to be achieved. In other words, such $f(0)$, $a_{kn}$, and $b_{kn}$ should be found so that the proposed function $f(x)$ models the dataset with minimum error. For this, the least-squares approach is utilized [38–40].

## 3. Numerical Results

With a linear regression study, it was aimed to forecast and model the passenger load factor (PLF) by using the information on the 2015–2016 year reservations, group sales data, calendar information, weekly dates, and trend difference between the current year and previous year, and load factor information of the same period of the previous year of Turkish Airlines. To cover all flying points, we chose 19 market areas to display our modeling results. All analysis and reports for linear modeling have been developed with the R programming language for portability and with Matlab for Fractional Model-3 and DAM and DAM wd modeling. In this section, we will discuss the numerical results of our four models.

The linear regression model gives meaningful results when looking at $R^2$ values as shown in Table 6 below. The high variability is 60%, which is seen in the aviation sector and above that $R^2$ (the coefficient of determination) gives consolidated results for the 30-day

forecasts. The model does not explain the uncertainty very well for the flights with 120 days or more remaining before their departures.

**Table 6.** Model Results.

| | RES 1 | RES 2 | RES 3 | RES 4 | RES 5 | RES 6 | RES 7 | RES 8 | RES 9 | RES 10 | RES 11 | RES 12 | RES 13 | RES 14 |
|---|---|---|---|---|---|---|---|---|---|---|---|---|---|---|
| Adjusted $R^2$ | 0.914 | 0.871 | 0.840 | 0.816 | 0.761 | 0.727 | 0.688 | 0.659 | 0.618 | 0.580 | 0.557 | 0.532 | 0.520 | 0.504 |

The model results were obtained using the 2016 data for testing. The mean absolute deviation statistic is used for model validation of the linear regression method. The mean absolute deviation (MAD) is a reliable measure of the variability of a single-variate sample of quantitative data in statistics. It can also refer to a population parameter generated from a sample and approximated by the MAD. Table 7 shows the validation findings for the machine learning–linear regression method.

**Table 7.** Model Validation MAD Data Frame.

| Area/Number of Days to Flight | 1 | 2 | 3 | 4 | 7 | 10 | 15 | 20 | 30 | 45 | 60 | 90 | 120 | 340 |
|---|---|---|---|---|---|---|---|---|---|---|---|---|---|---|
| Eastern Africa—Turkey | 0.02 | 0.02 | 0.06 | 0.05 | 0.05 | 0.06 | 0.08 | 0.09 | 0.1 | 0.08 | 0.07 | 0.09 | 0.09 | 0.1 |
| Far East—Turkey | 0.04 | 0.05 | 0.05 | 0.07 | 0.05 | 0.06 | 0.06 | 0.07 | 0.09 | 0.08 | 0.07 | 0.07 | 0.08 | 0.08 |
| Europe—Turkey | 0.03 | 0.05 | 0.08 | 0.09 | 0.08 | 0.09 | 0.1 | 0.1 | 0.09 | 0.09 | 0.09 | 0.09 | 0.09 | 0.09 |
| Middle East—Turkey | 0.05 | 0.08 | 0.08 | 0.09 | 0.07 | 0.09 | 0.09 | 0.09 | 0.09 | 0.1 | 0.1 | 0.1 | 0.09 | 0.09 |
| North Atlantic—Turkey | 0.01 | 0.03 | 0.05 | 0.07 | 0.05 | 0.06 | 0.08 | 0.11 | 0.11 | 0.08 | 0.09 | 0.07 | 0.07 | 0.06 |
| Northern Africa—Turkey | 0.05 | 0.06 | 0.08 | 0.09 | 0.07 | 0.07 | 0.08 | 0.07 | 0.07 | 0.1 | 0.05 | 0.08 | 0.07 | 0.07 |
| South Atlantic—Turkey | 0.01 | 0.03 | 0.05 | 0.05 | 0.07 | 0.07 | 0.08 | 0.07 | 0.07 | 0.1 | 0.05 | 0.08 | 0.07 | 0.07 |
| Southern Africa—Turkey | 0.01 | 0.02 | 0.04 | 0.02 | 0.02 | 0.02 | 0.08 | 0.13 | 0.13 | 0.12 | 0.09 | 0.08 | 0.08 | 0.07 |
| Turkey—Eastern Africa | 0.03 | 0.04 | 0.03 | 0.07 | 0.07 | 0.12 | 0.06 | 0.1 | 0.09 | 0.08 | 0.08 | 0.10 | 0.09 | 0.09 |
| Turkey—Far East | 0.03 | 0.04 | 0.05 | 0.06 | 0.07 | 0.1 | 0.08 | 0.08 | 0.09 | 0.09 | 0.08 | 0.09 | 0.09 | 0.09 |
| Turkey—Europe | 0.04 | 0.05 | 0.05 | 0.07 | 0.07 | 0.09 | 0.08 | 0.08 | 0.09 | 0.09 | 0.09 | 0.09 | 0.09 | 0.09 |
| Turkey—Middle East | 0.04 | 0.06 | 0.08 | 0.08 | 0.08 | 0.09 | 0.11 | 0.11 | 0.1 | 0.1 | 0.1 | 0.1 | 0.1 | 0.09 |
| Turkey—North Atlantic | 0.04 | 0.02 | 0.03 | 0.04 | 0.05 | 0.08 | 0.06 | 0.06 | 0.07 | 0.08 | 0.07 | 0.07 | 0.06 | 0.06 |
| Turkey—Northern Africa | 0.05 | 0.06 | 0.07 | 0.1 | 0.07 | 0.09 | 0.11 | 0.09 | 0.11 | 0.11 | 0.11 | 0.12 | 0.1 | 0.09 |
| Turkey—South Atlantic | 0.03 | 0.04 | 0.05 | 0.09 | 0.06 | 0.07 | 0.05 | 0.08 | 0.11 | 0.11 | 0.11 | 0.12 | 0.1 | 0.09 |
| Turkey—Southern Africa | 0.02 | 0.03 | 0.05 | 0.02 | 0.06 | 0.1 | 0.03 | 0.08 | 0.11 | 0.08 | 0.07 | 0.09 | 0.09 | 0.08 |
| Turkey—Turkey | 0.08 | 0.08 | 0.08 | 0.09 | 0.09 | 0.09 | 0.08 | 0.09 | 0.08 | 0.09 | 0.09 | 0.08 | 0.08 | 0.08 |
| Turkey—Western Africa | 0.05 | 0.04 | 0.07 | 0.09 | 0.1 | 0.11 | 0.08 | 0.08 | 0.11 | 0.09 | 0.09 | 0.10 | 0.09 | 0.09 |
| Western Africa—Turkey | 0.03 | 0.03 | 0.07 | 0.06 | 0.07 | 0.06 | 0.08 | 0.08 | 0.08 | 0.09 | 0.08 | 0.08 | 0.09 | 0.08 |

The Fractional Model-3, DAM, and DAM wd modeling results were obtained using the 2016 reservation rate data (real load factor data) to find the optimum modeling method. We used modeling days as RES0, RES1, RES2, RES3, RES4, RES5, RES6, RES7, RES8, RES9, RES10, RES12, RES15, RES20, RES30, RES45, RES60, RES90, RES120, RES200, and RES340 days to flight. We have used the same *M* and *l* constants for the comparison.

For model validation, the mean absolute percentage error (MAPE) is utilized. MAPE is a reliable measure of the variability of quantitative data samples in statistics. In Table 8, MAPE and optimum $\alpha$ value results for Fractional Model-3 are given. For *N* = 15, we have found the least AMAPE value to be 0.734.

Then, we used the DAM model and found the results for different *M* and *l* values. We have found the least AMAPE value to be 0.853 for *M* = 7 and *l* = 3, as shown in Table 9.

Then, we used the DAM wd model, which is the first-order derivative-added version of the DAM model, and found the results for different *M* and *l* values. We have found the least AMAPE value to be 0.571 for *M* = 7 and *l* = 3, as shown in Table 10.

**Table 8.** Fractional Model-3 modeling MAPE and optimum $\alpha$ value results.

| Marketing Area | Fractional Model-3 (N = 5) | | Fractional Model-3 (N = 9) | | Fractional Model-3 (N = 15) | |
|---|---|---|---|---|---|---|
| | MAPE | $\alpha$ Value | MAPE | $\alpha$ Value | MAPE | $\alpha$ Value |
| Eastern Africa—Turkey | 2.26 | 0.02 | 1.03 | 0.77 | 0.47 | 0.81 |
| Far East—Turkey | 2.49 | 0.96 | 0.92 | 0.54 | 0.70 | 0.99 |
| Europe—Turkey | 1.10 | 0.74 | 0.63 | 0.35 | 0.21 | 0.67 |
| Middle East—Turkey | 3.56 | 0.01 | 1.74 | 0.27 | 0.49 | 0.51 |
| North Atlantic—Turkey | 3.14 | 0.44 | 0.38 | 0.44 | 0.30 | 0.50 |
| Northern Africa—Turkey | 5.06 | 0.50 | 1.87 | 0.27 | 0.74 | 0.52 |
| South Atlantic—Turkey | 5.60 | 0.01 | 2.62 | 0.30 | 1.88 | 0.95 |
| Southern Africa—Turkey | 0.91 | 0.01 | 0.51 | 0.82 | 0.48 | 0.46 |
| Turkey—Northern Africa | 5.07 | 0.45 | 2.33 | 0.08 | 1.59 | 0.65 |
| Turkey—Far East | 3.65 | 1.00 | 0.70 | 0.68 | 0.34 | 0.51 |
| Turkey—South Atlantic | 3.84 | 0.25 | 1.48 | 1.00 | 0.65 | 0.75 |
| Turkey—Europe | 3.62 | 0.67 | 1.35 | 0.65 | 1.16 | 0.56 |
| Turkey—Western Africa | 7.29 | 0.11 | 2.33 | 0.30 | 0.94 | 0.95 |
| Turkey—Eastern Africa | 5.40 | 0.12 | 1.14 | 0.92 | 0.72 | 1.00 |
| Turkey—North Atlantic | 6.54 | 0.17 | 1.80 | 0.35 | 0.63 | 0.73 |
| Turkey—Middle East | 3.23 | 0.02 | 1.03 | 0.37 | 0.42 | 0.83 |
| Turkey—Southern Africa | 5.36 | 0.01 | 1.20 | 0.82 | 0.43 | 0.52 |
| Western Africa—Turkey | 2.57 | 0.41 | 1.97 | 0.31 | 0.92 | 0.44 |
| Turkey—Turkey | 4.39 | 0.01 | 1.70 | 0.31 | 0.87 | 0.87 |
| AMAPE | 3.951 | | 1.407 | | 0.734 | |

**Table 9.** DAM modeling MAPE and optimum $\gamma$ value results.

| Marketing Area | DAM (M = 3 and l = 3) | | DAM (M = 5 and l = 3) | | DAM (M = 7 and l = 3) | |
|---|---|---|---|---|---|---|
| | MAPE | $\gamma$ Value | MAPE | $\gamma$ Value | MAPE | $\gamma$ Value |
| Eastern Africa—Turkey | 1.25 | 0.92 | 0.64 | 0.67 | 0.53 | 0.91 |
| Far East—Turkey | 2.28 | 0.99 | 0.72 | 0.25 | 0.75 | 0.43 |
| Europe—Turkey | 0.51 | 0.03 | 0.48 | 0.25 | 0.36 | 0.49 |
| Middle East—Turkey | 1.20 | 0.98 | 0.71 | 0.64 | 0.53 | 0.91 |
| North Atlantic—Turkey | 0.89 | 0.90 | 0.42 | 0.68 | 0.35 | 0.43 |
| Northern Africa—Turkey | 1.26 | 0.92 | 1.07 | 0.76 | 0.88 | 0.91 |
| South Atlantic—Turkey | 2.65 | 0.92 | 2.52 | 0.15 | 1.86 | 0.43 |
| Southern Africa—Turkey | 0.61 | 0.95 | 0.60 | 0.14 | 0.46 | 0.43 |
| Turkey—Northern Africa | 2.03 | 0.96 | 1.86 | 0.89 | 1.77 | 0.43 |
| Turkey—Far East | 2.37 | 0.90 | 0.78 | 0.68 | 0.51 | 0.43 |
| Turkey—South Atlantic | 3.39 | 0.98 | 1.83 | 0.92 | 0.93 | 0.36 |
| Turkey—Europe | 1.91 | 0.01 | 1.47 | 0.92 | 1.18 | 0.49 |
| Turkey—Western Africa | 4.68 | 0.90 | 2.17 | 0.96 | 1.07 | 0.01 |
| Turkey—Eastern Africa | 3.17 | 0.90 | 2.30 | 0.89 | 0.88 | 0.01 |
| Turkey—North Atlantic | 2.62 | 0.94 | 1.35 | 0.99 | 0.59 | 0.91 |
| Turkey—Middle East | 2.74 | 0.98 | 0.94 | 0.64 | 0.66 | 0.43 |
| Turkey—Southern Africa | 3.01 | 0.90 | 1.65 | 0.76 | 1.02 | 0.43 |
| Western Africa—Turkey | 1.88 | 0.03 | 1.12 | 0.26 | 0.63 | 0.34 |
| Turkey—Turkey | 2.22 | 0.05 | 1.51 | 0.92 | 1.24 | 0.43 |
| AMAPE | 2.14 | | 1.270 | | 0.853 | |

In Tables 8–10, we have listed 19 market route areas of Turkish Airlines—for instance, in eastern Africa–Turkey; the departure area is eastern Africa and the arrival area is Turkey. First, Fractional Model-3 was used in this work to compare different *N* exponent values up to 5, 9, and 15 (*N* = 5, 9, 15) in Equation (5); we have found MAPE and $\alpha$ value results as shown in Table 8. For *N* = 15, we have found the most accurate results that are approximately the same with the real reservation rate data. Then, we used another model, which is DAM (deep assessment methodology) for different *M* and *l* values that are chosen arbitrarily to find the optimum $\alpha$ values that correspond to the least MAPE values for the reservation rate data and also DAM wd, which is the first-order derivative-added version

of the DAM model. Here, $M$ defines how many truncations are performed, and the $l$ value shows how many values to model every time. For instance, using value $l = 2$, it means that the data between 340 days to flight and boarding days were modeled using data from the previous two values every time. Also, our Matlab code changes $\alpha$ values between 0 and 1 automatically in seconds to find the optimum modeling results, as shown by the MAPE values in Tables 8–10 provided below.

**Table 10.** DAM wd modeling MAPE and optimum $\gamma$ value results.

| Marketing Area | DAM wd (M = 3 and l = 3) | | DAM wd (M = 5 and l = 3) | | DAM wd (M = 7 and l = 3) | |
|---|---|---|---|---|---|---|
| | MAPE | $\gamma$ Value | MAPE | $\gamma$ Value | MAPE | $\gamma$ Value |
| Eastern Africa—Turkey | 1.43 | 0.99 | 0.51 | 0.02 | 0.40 | 0.08 |
| Far East—Turkey | 2.54 | 0.99 | 0.75 | 0.95 | 0.42 | 0.13 |
| Europe—Turkey | 0.53 | 0.24 | 0.53 | 0.90 | 0.30 | 0.45 |
| Middle East—Turkey | 0.88 | 0.01 | 0.65 | 0.02 | 0.34 | 0.30 |
| North Atlantic—Turkey | 1.03 | 0.99 | 0.40 | 0.90 | 0.31 | 0.71 |
| Northern Africa—Turkey | 1.32 | 0.97 | 0.94 | 0.93 | 0.65 | 0.51 |
| South Atlantic—Turkey | 2.29 | 0.03 | 0.69 | 0.06 | 0.61 | 0.76 |
| Southern Africa—Turkey | 0.60 | 0.03 | 0.41 | 0.67 | 0.42 | 0.68 |
| Turkey—Northern Africa | 2.10 | 0.76 | 1.85 | 0.01 | 0.79 | 0.28 |
| Turkey—Far East | 2.66 | 0.99 | 0.75 | 0.90 | 0.23 | 0.76 |
| Turkey—South Atlantic | 3.24 | 0.99 | 1.50 | 0.93 | 0.93 | 0.15 |
| Turkey—Europe | 1.97 | 0.99 | 1.46 | 0.94 | 0.90 | 0.34 |
| Turkey—Western Africa | 2.70 | 0.99 | 1.20 | 0.94 | 0.96 | 0.77 |
| Turkey—Eastern Africa | 2.93 | 0.97 | 0.91 | 0.94 | 0.70 | 0.81 |
| Turkey—North Atlantic | 2.31 | 0.99 | 1.16 | 0.94 | 0.66 | 0.91 |
| Turkey—Middle East | 2.26 | 0.99 | 1.03 | 0.99 | 0.46 | 0.17 |
| Turkey—Southern Africa | 3.41 | 0.99 | 1.39 | 0.94 | 0.52 | 0.74 |
| Western Africa—Turkey | 1.23 | 0.97 | 1.13 | 0.94 | 0.66 | 0.19 |
| Turkey—Turkey | 2.43 | 0.99 | 0.77 | 0.04 | 0.62 | 0.74 |
| AMAPE | 1.99 | | 0.948 | | 0.571 | |

The models were compared using the mean absolute percentage error (MAPE). The MAPE formulation is shown in Equation (17).

$$MAPE = \frac{1}{k}\sum_{i=1}^{k}\left|\frac{p(i) - \widetilde{p}(i)}{p(i)}\right| \times 100 \tag{17}$$

where $p(i)$ is the real value, and $\widetilde{p}(i)$ is the predicted value.

DAM and the fractional model are theoretically identical when the fractional order ($\alpha$) value in the fractional model equals one. The alpha values in the fractional model were taken between 0.001 and 1 and increased by 0.001. As a result, some results in the DAM model and the fractional model may be the same. The minimal MAPE values were used to determine the alpha values.

Tables 8–10 show MAPE values of the load factor versus the number of days to flight according to the methods of fractional calculus, DAM, and DAM wd model. Considering Table 8, when the truncation number $N$ in Equation (5) was increased, the MAPE ratio in fractional models was decreased as expected. The average of the total MAPE (AMAPE) was calculated with the formula given in

$$AMAPE = \frac{\sum MAPE}{M} \tag{18}$$

where $M$ represents the number of values, which was 19.

To compare the performance of the DAM model, DAM wd model, and Fractional Model-3, we calculated the MAPE ratio for each. The MAPE of each DAM model was divided by the corresponding Fractional Model-3's MAPE, and the maximum and minimum

values were established as benchmarks. The DAM model's MAPE results were 0.68 to 2.39 times smaller than the fractional model's MAPE. The DAM wd model's MAPE results were 0.33 to 1108 times smaller than the DAM model's MAPE. Approximate values were obtained for Fractional Model-3 ($N = 15$) and DAM model ($M = 7$ and $l = 3$). When N equaled 9, the fractional model's AMAPE was 1.407, while the DAM model ($M = 3$ and $l = 3$) had an AMAPE of 2.14. For DAM ($M = 5$ and $l = 3$), the results were about 1.16 times better than the fractional model $N = 9$. At N = 5, the fractional model's AMAPE was 3.951, and the DAM model ($M = 3$ and $l = 3$) reported an AMAPE of 2.14. The DAM wd model ($M = 7$ and $l = 3$) outperformed the fractional model ($N = 15$) by about 1.43 times in the highest AMAPE values.

At $N = 9$ and $N = 15$, the fractional model surpassed the DAM model for DAM ($M = 3$ and $l = 3$). Using the fractional model, DAM model, and DAM wd model yielded better outcomes with fewer computational steps. In other words, the DAM and DAM wd models achieve the same MAPE with fewer terms compared to the Fractional Model-3. DAM wd ($M = 7$ and $l = 3$) had the lowest AMAPE value at 0.571, superior to DAM wd ($M = 3$ and $l = 3$) and DAM wd ($M = 5$ and $l = 3$). Lower AMAPE values were evident compared to the DAM and Fractional Model-3 modeling.

To visualize our modeling results, we acquired results using linear regression (machine learning), Fractional Model-3, DAM wd, and DAM model, respectively. We compared the modeling results of four modeling methods with real reservation rates that are shown in Figures 1–4. It is clear that until 120 days before the flight, linear regression gives mostly the results of the previous year's load factor rate to this year because the reservation pattern is not obtained regularly. After 120 days, with the increment of the reservation rates, we can find more accurate results for the load factor. Because of that, we obtain the more accurate results that are shown in our figures for 120 days to the boarding day.

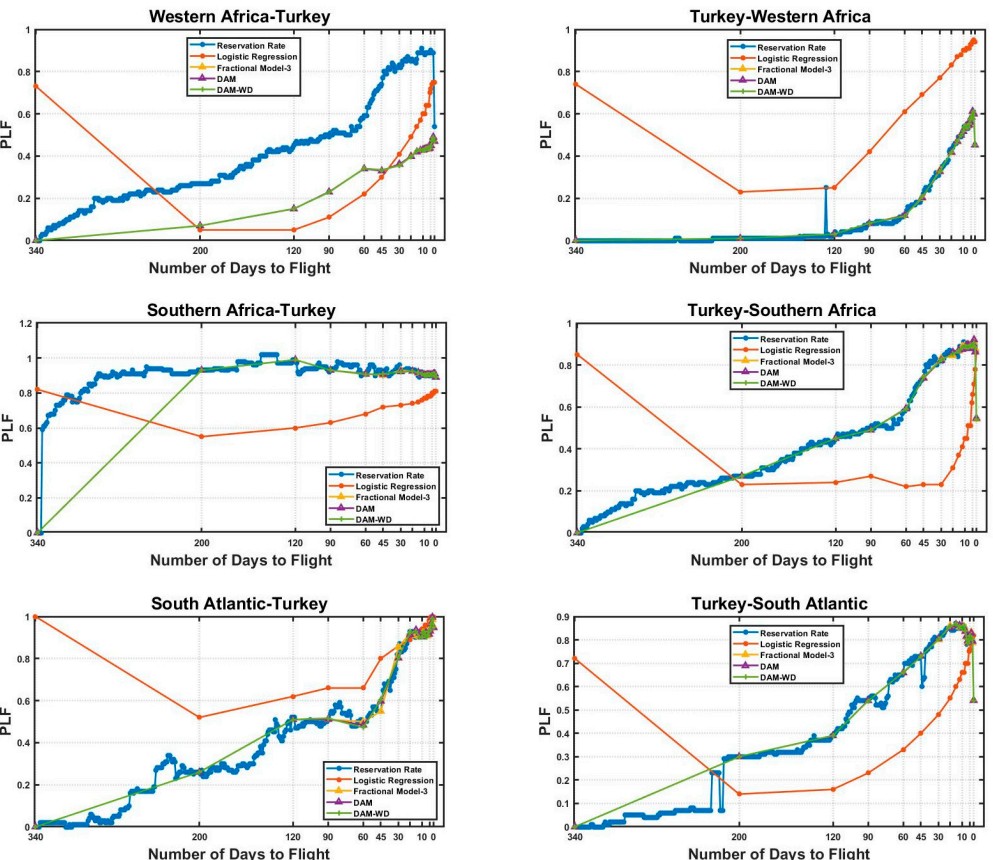

**Figure 1.** Model Results of DAM wd, DAM, Fractional Model-3, Linear Regression, and reservation rates of the year 2016 for Western Africa, Southern Africa, and South Atlantic.

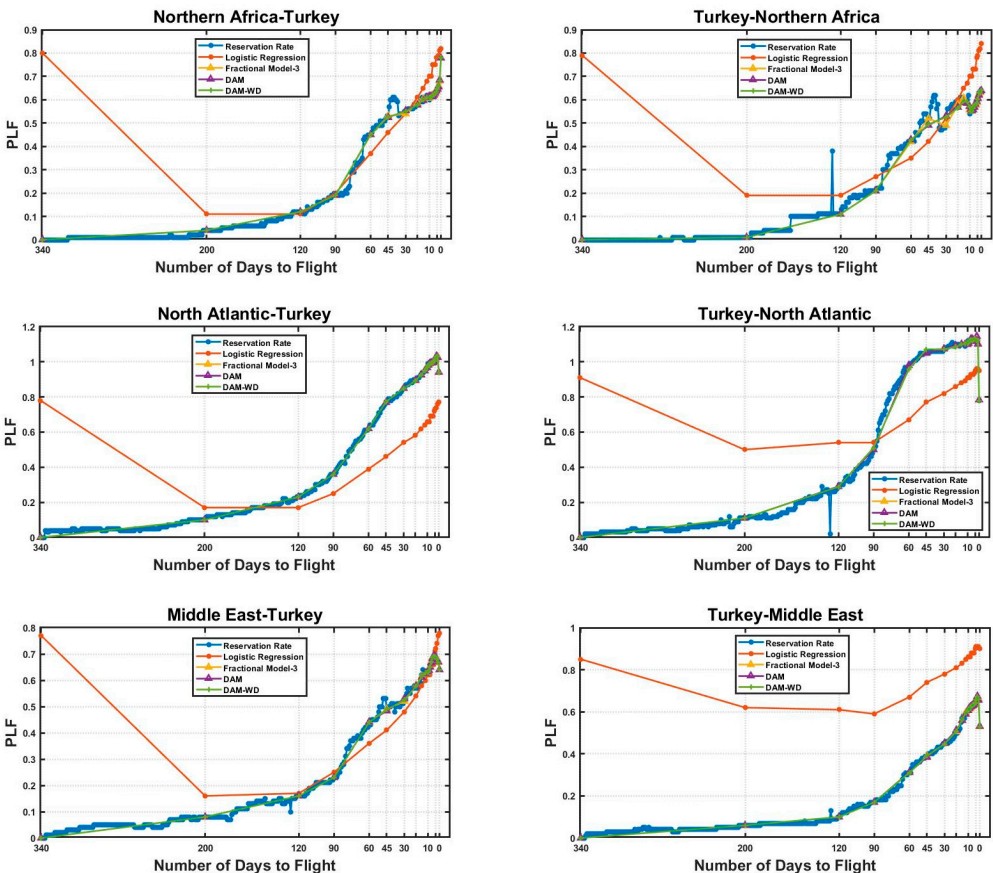

**Figure 2.** Model Results of DAM wd, DAM, Fractional Model-3, Linear Regression and reservation rates of the year 2016 for Northern Africa, North Atlantic, and Middle East.

Figures 1–4 depict how the load factor value varies depending on the number of days until the flight in various locations. The last point on the graph, read from right to left, provides us the load factor value of the boarding day. Flights to market areas like the North Atlantic, South Atlantic, Far East, and South Africa, especially when considering the distance, demonstrate that planes are starting to fill up, even if they have more time to take off. We can easily see when flights from Turkey to Turkey, that is, domestic flights, begin to fill up, especially recently. Flights to different destinations exhibit different patterns, as may be observed from these inferences. For capturing these trends, the panel data format is critical. The vertical lines demonstrate how the reservation days are subject to change. If we look at the South Atlantic–Turkey destination, for example, each booking day has very varying values. As a result, forecasting will be difficult, and variations will be significant. DAM wd, DAM, and Fractional Model-3 provide similar modeling results. The linear regression method gives more accurate results for 340 days to flight because it uses the previous year's reservation rates and other factors like day of the week, etc. Before 200 days to flight, four modeling methods obtained similar differences with real reservation data. While the departure day is soon, when more reservation data are included in the dataset, more accurate modeling results are obtained. Especially in the last 30 days to departure, all methods have more approximate modeling results, although there are cancellations and reservation changes.

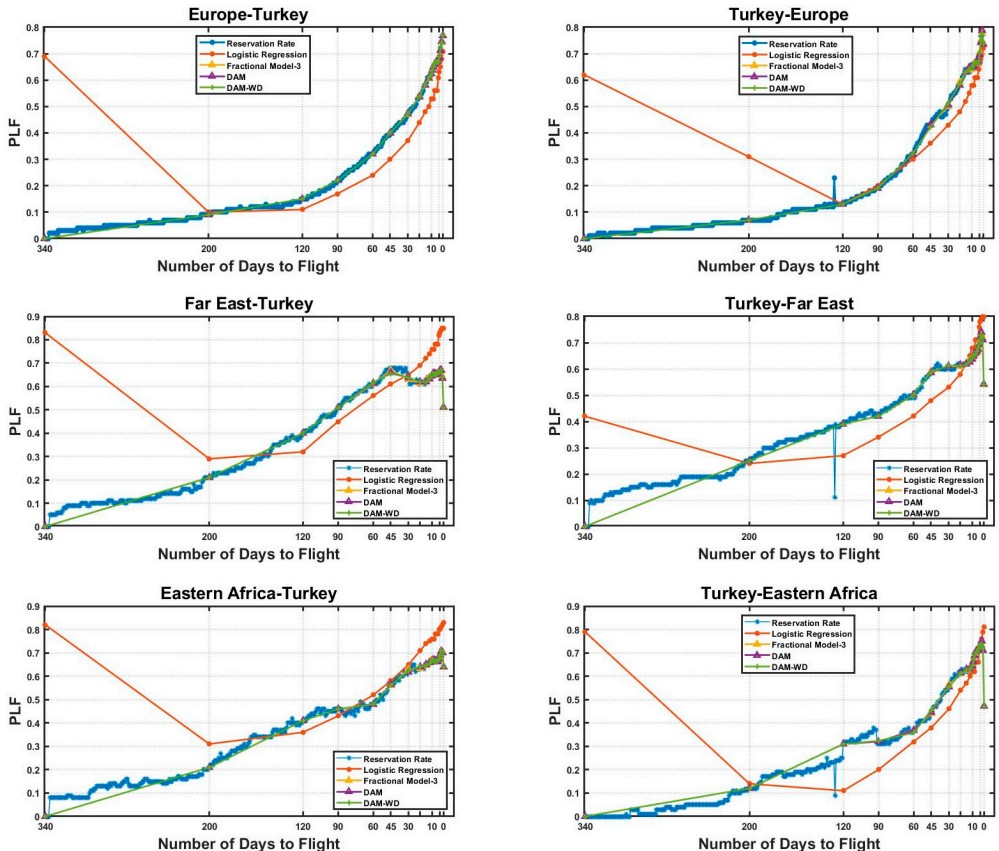

**Figure 3.** Model Results of DAM wd, DAM, Fractional Model-3, Linear Regression, and reservation rates of the year 2016 for Europe, Far East, and Eastern Africa.

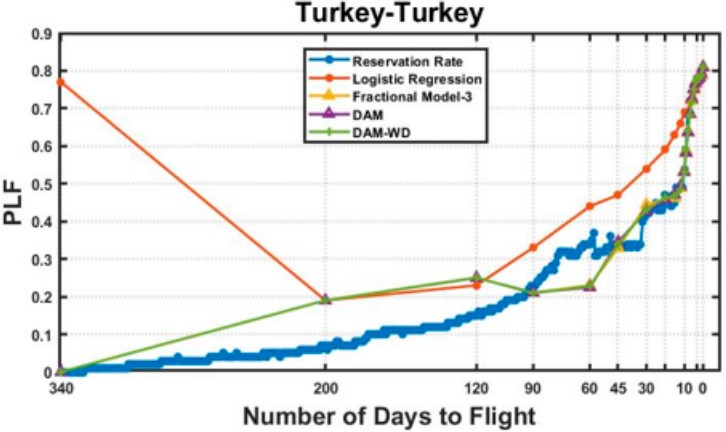

**Figure 4.** Model Results of DAM wd, DAM, Fractional Model-3, Linear Regression, and reservation rates of year 2016 for Turkey's Domestic Flights.

For the Northern Africa–Turkey, Turkey–Northern Africa, North Atlantic–Turkey, Turkey–North Atlantic, Middle East–Turkey, and Turkey–Middle East market areas, Fractional Model-3 (yellow line), DAM (purple line), and DAM wd results (green line) are more approximate to real reservation rates than the linear regression (red line) modeling results, as shown in Figure 2.

For the Europe–Turkey and Turkey–Europe market areas, the DAM results (yellow line) are more approximate to real reservation rates than the linear regression (red line) modeling results, as shown in Figure 3. In the Far East–Turkey market area after 30 days to flight and for Turkey–Far East after 15 days to flight, the linear regression (red line)

modeling results are more approximate than the DAM modeling results. For Eastern Africa–Turkey after 80 days to flight and for Turkey–Eastern Africa after 18 days to flight, the linear regression (red line) modeling results were more approximate to real reservation rates than the DAM modeling results.

For domestic flights within Turkey, the DAM results (yellow line) and DAM wd results (green line) are more approximate to real reservation rates than the linear regression (red line) modeling results, especially 30 days to the departure date, as shown in Figure 4.

The DAM wd technique was used to model real reservation flight data with three distinct $M$ and $l$ values in the following phase because it provides the most approximate modeling results among the four modeling approaches (Table 11). We used the same constants to model market areas. For market areas and sample flights, the modeling outcomes using DAM wd with the same constants are quite similar. The lowest MAPE value was the 0.11 obtained for $M = 7$ and $l = 3$ for flight number 45. Predicting flight results is challenging due to varying values on each booking day. Linear regression predominantly relies on the previous year's load factor until 120 days before the flight, with improved accuracy after that due to rising reservation rates. Different flight markets exhibit distinct patterns; for instance, flights from Turkey to Turkey show increasing demand when the departure day is soon. DAM wd modeling the least MAPE results of ($M = 3$ and $l = 3$) is 0.44, ($M = 5$ and $l = 3$) is 0.27, and ($M = 7$ and $l = 3$) is 0.11, based on data from 32 sample flights. In African flights, the MAPE values are high because the reservation rates are unstable, especially 200 days before flights. Far East and Europe flights have smaller MAPE values.

**Table 11.** DAM wd modeling MAPE and optimum γ value results for sample flights.

| Marketing Area | Flight Number | DAM wd (M = 3 and l = 3) | | DAM wd (M = 5 and l = 3) | | DAM wd (M = 7 and l = 3) | |
|---|---|---|---|---|---|---|---|
| | | MAPE | γ Value | MAPE | γ Value | MAPE | γ Value |
| Eastern Africa—Turkey | 677 | 1.59 | 0.45 | 1.57 | 0.95 | 0.6 | 0.05 |
| | 25 | 1.42 | 0.59 | 1.4 | 0.99 | 1.16 | 0.83 |
| Far East—Turkey | 47 | 1.6 | 0.99 | 1.17 | 0.99 | 0.67 | 0.86 |
| | 717 | 2.12 | 0.97 | 1.66 | 0.02 | 1.11 | 0.35 |
| | 731 | 0.68 | 0.97 | 0.68 | 0.98 | 0.55 | 0.27 |
| Europe—Turkey | 965 | 2.12 | 0.03 | 1.79 | 0.09 | 1.84 | 0.97 |
| | 1618 | 0.83 | 0.78 | 0.39 | 0.01 | 0.38 | 0.01 |
| Middle East—Turkey | 811 | 1.23 | 0.03 | 1.18 | 0.95 | 0.8 | 0.05 |
| | 4389 | 0.84 | 0.78 | 0.4 | 0.01 | 0.38 | 0.01 |
| North Atlantic—Turkey | 78 | 0.73 | 0.01 | 0.62 | 0.02 | 0.6 | 0.04 |
| | 82 | 1.07 | 0.01 | 0.82 | 0.96 | 0.65 | 0.04 |
| Northern Africa—Turkey | 618 | 2.5 | 0.03 | 2.06 | 0.02 | 2.03 | 0.92 |
| South Atlantic—Turkey | 16 | 0.44 | 0.97 | 0.44 | 0.95 | 0.25 | 0.07 |
| Southern Africa—Turkey | 43 | 0.78 | 0.45 | 0.53 | 0.02 | 0.52 | 0.04 |
| | 45 | 0.39 | 0.01 | 0.27 | 0.03 | 0.11 | 0.13 |
| Turkey-Eastern Africa | 676 | 1.98 | 0.01 | 0.76 | 0.79 | 0.47 | 0.05 |
| Turkey—Far East | 62 | 0.6 | 0.45 | 0.48 | 0.02 | 0.32 | 0.04 |
| | 64 | 2.24 | 0.45 | 1.1 | 0.02 | 0.45 | 0.06 |
| Turkey-Europe | 1353 | 0.5 | 0.01 | 0.36 | 0.02 | 0.14 | 0.63 |
| Turkey—Middle East | 140 | 2.05 | 0.01 | 1.66 | 0.9 | 0.58 | 0.06 |
| | 7 | 1.23 | 0.99 | 0.95 | 0.02 | 0.27 | 0.18 |
| Turkey—North Atlantic | 17 | 0.55 | 0.03 | 0.53 | 0.01 | 0.54 | 0.89 |
| | 33 | 0.99 | 0.54 | 0.71 | 0.17 | 0.64 | 0.12 |
| | 663 | 3.34 | 0.01 | 3.21 | 0.02 | 2.65 | 0.04 |
| Turkey—Northern Africa | 680 | 3.03 | 0.45 | 2.5 | 0.02 | 1.61 | 0.04 |
| | 690 | 4.77 | 0.01 | 1.93 | 0.93 | 1.23 | 0.06 |
| Turkey—Southern Africa | 42 | 0.71 | 0.01 | 0.69 | 0.99 | 0.28 | 0.83 |
| | 44 | 0.83 | 0.01 | 0.45 | 0.02 | 0.36 | 0.83 |
| | 589 | 2.01 | 0.01 | 1.34 | 0.01 | 1.29 | 0.04 |
| Turkey—Western Africa | 623 | 1.96 | 0.01 | 1.35 | 0.96 | 1.37 | 0.6 |
| | 625 | 1.86 | 0.99 | 1.81 | 0.97 | 1.78 | 0.99 |
| Turkey—Middle East | 626 | 2.24 | 0.54 | 1.35 | 0.02 | 1.27 | 0.83 |

## 4. Conclusions

We propose a method for passenger load factor (PLF) prediction that involves modeling the relationship between the load factor and the number of days remaining until a flight.

Our approach yields minimal errors compared to other methods like DAM, fractional calculus, and standard linear methods. We leveraged data from 19 Turkish Airlines market routes and sample flights to construct a continuous curve applicable to any time frame using the least-squares approach, incorporating fractional calculus theory and a linear model. By utilizing historical data from the reservation process development, our method enables the anticipation of future flight values.

The analysis of the DAM wd model using specific coefficients demonstrates superior performance compared to linear techniques and Fractional Model-3 and DAM modeling. In this paper, we compare and contrast the effectiveness of the deep assessment system (DAM) with the first-order derivative and DAM models in simulating air transport PLF, Fractional Model-3, and linear regression method results. We conclude that the DAM model with a first-order derivative excels not only in modeling capabilities but also in its ability to predict such data accurately.

Our provided approaches allow the extraction of load factor development parameters for any chosen time using the DAM method and discrete data points for each percentile, resulting in a more precise continuous curve. Tables 8–10 display load factor modeling outcomes for linear, DAM, and DAM wd models for three different constant values.

Each of the DAM, DAM wd, and Fractional Model-3 models exhibit a decrease in the MAPE ratio as the exponent number in Equation (5) rises. When examining the $M$ and $l$ values for the DAM model, we observe that altering the $M$ value has a greater impact on results than changing the l value.

The modeling findings with the DAM wd model are approximately 0.67 times superior to the DAM model and outperform the fractional model and regression analysis. For the DAM wd modeling method, which models real reservation data with varying $M$ and $l$ values, the lowest AMAPE value is 0.571. In practical reservation modeling, each booking day shows considerable variation, making forecasting challenging. Our AMAPE results for DAM wd modeling are as follows: ($M = 3$ and $l = 3$) 1.99, ($M = 5$ and $l = 3$) 0.948, and ($M = 7$ and $l = 3$) 0.571. These values are lower than the DAM modeling results, which are ($M = 3$ and $l = 3$) 2.14, ($M = 5$ and $l = 3$) 1.27, and ($M = 7$ and $l = 3$) 0.853. A detailed analysis reveals that the linear regression method yields less accurate estimations during the modeling phase. We developed a load factor prediction model to enhance the management of Turkish Airlines' flight capacity. As we opted for a dynamic and conditional model approach, the models are trained daily through parallel processing tools. Over a 30-day forecast, our model predicts flight results with an error range of 0 to 8%, accurately predicting around 64% of the flights.

This study's limitation revolves around the predictive accuracy of the proposed method beyond a 30-day forecast for reservations, where deviations greater than 10% may occur. This highlights a potential challenge in achieving reliable long-term forecasting accuracy. To address this limitation, future research could focus on refining the modeling approach to enhance its predictive capability for extended forecasting periods. Possible remedies for improvement may include exploring advanced forecasting techniques, incorporating additional variables or factors into the model, proposing the DAM with first and second derivatives, and optimizing the methodology to make it more robust and adaptable to longer-term prediction scenarios. By addressing this limitation, future studies could enhance the practical applicability and reliability of the proposed method in real-world air transport capacity management contexts.

To seamlessly integrate our proposed model into the operational system, flight accuracy must be further optimized. Our forecasting model, including factors like special day effects, significantly outperforms the prior linear model. Revenue and capacity management in air transport are increasingly embracing data-driven optimization methods. Thus, there is a growing need for efficient models grounded in extensive available data to address these traditional challenges. Our ongoing effort focuses on developing an infrastructure to handle sales days separately from reservations. We plan to overcome this limitation by incorporating emerging technologies such as distributed computing or cloud

computing to process data. Our emphasis lies in adopting a novel fractional calculus approach to manage vast amounts of data by forecasting reservation intervals and enhancing model performance.

We also intend to incorporate the inflation rate variable into the model as an extension of this study's scope for an improved multi-input model. Additionally, using the model for daily and monthly evaluations, along with integrating second- and higher-order derivatives, would enhance our DAM wd model. While our current study assessed individual flights and marketing areas, future research will explore how PLF factors of one marketing area or flight affect others and to what extent changes in one area or flight influence another.

**Author Contributions:** The contribution of each author is listed as follows: E.K. has contributed to supervision, conceptualization, investigation, methodology, and administration. V.T. plays an important role in resources, supervision, and validation. K.Ş. was the key person in investigation, administration, validation, and writing. K.K. supported conceptualization, writing, and editing. N.Ö.Ö.T. has contributed to validation, writing, and editing. All authors have read and agreed to the published version of the manuscript.

**Funding:** This work was supported by the Vodafone Future Laboratory, Istanbul Technical University (ITU), under Grant ITUVF20180901P11.

**Data Availability Statement:** The data, R, and Matlab scripts used to support the findings of this study are available from the corresponding author upon request.

**Acknowledgments:** We would like to express our sincere gratitude to Turkish Airlines Inc. for the data for this study that were processed and graciously provided to us. Any thoughts, findings, conclusions, or recommendations stated in this material are solely those of the author(s) and do not necessarily reflect Turkish Airlines' view. This work was supported by the Scientific Research Projects Department of Istanbul Technical University, Project Number MGA-2021-43392.

**Conflicts of Interest:** The authors declare no conflicts of interest. The funders had no role in the design of the study, in the collection, analyses or interpretation of data, in the writing of the manuscript, or in the decision to publish the results.

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
