# Peer review of "Modeling and Predicting Passenger Load Factor in Air Transportation: A Deep Assessment Methodology with Fractional Calculus Approach Utilizing Reservation Data"

_fractalfract, doi:10.3390/fractalfract8040214_

Round 1

Reviewer 1 Report

Comments and Suggestions for Authors

Paper deals with important task. It has a scientific novelty. It has a logical structure. The paper is technically sound. The experimental section is good. The proposed approach is logical, results are clear.

Suggestions: 

1.    It would be good to add the remainder of this paper.

2.    The significance of this study needs to be better justified. Its role for practice, theory and future research.

3.    The authors do not substantiate why the dynamics of the level of bookings of Turkish airlines over time can be described by means of the ratio (5)? Why is it that the dynamics of the level of bookings of Turkish airlines over time will have a fractal rather than traditional character?

4.    It is necessary to describe the dataset directly used for linear regression training in more detail.

5.    Panel Data contains categorical values. The article does not clearly describe how categorical data are converted to numerical data and why this particular method is optimal

6.    It would be appropriate to expand the overview of the application of the fractal approach in mathematical modeling. For example, pay attention to publications DOI: 10.3390/fractalfract7080584, DOI: 10.1016/j.rinp.2020.103702 and DOI: 10.1007/978-3-031-04812-8_9

7.    Conclusion section should be extended using: 

·       limitations of the proposed approach;

·       prospects for future research.

Author Response

Response to Review 1

First, we would like to express our gratitude to the valuable Reviewer since without his/her remarks, the article would not be complete. Thank you, a lot. We will try to answer in order.

  1. It would be good to add the remainder of this paper.
  1. The significance of this study needs to be better justified. Its role for practice, theory and future research.

Abstract and other parts are changed and improved.

  1. The authors do not substantiate why the dynamics of the level of bookings of Turkish airlines over time can be described by means of the ratio (5)? Why is it that the dynamics of the level of bookings of Turkish airlines over time will have a fractal rather than traditional character?

While the number of reservations progresses linearly until 120 days before the flight, reservation data takes a nonlinear structure after 120 days, especially due to group passenger cancellations and aircraft type changes, etc. For this reason, the fractal approach was adopted in our study.

Fractal analysis offers several advantages over linear regression:

Capturing Complex Patterns: Fractal analysis can capture complex, nonlinear patterns in data, whereas linear regression assumes a linear relationship between variables, which may not be suitable for all datasets.

Scale-Invariance: Fractal analysis accounts for scale-invariance, meaning that patterns observed at different scales are analyzed consistently. Linear regression may not capture the intricate details of data across various scales.

Robustness to Outliers: Fractal analysis techniques are often more robust to outliers compared to linear regression, which can be heavily influenced by extreme data points.

Model Flexibility: Fractal analysis allows for the exploration of self-similarity and self-affinity in data, providing insights into complex structures that linear regression may overlook due to its rigid linear assumptions.

Applicability to Nonlinear Systems: Fractal analysis is particularly useful for analyzing nonlinear systems and processes, where linear regression may not provide accurate or meaningful results.

Overall, fractal analysis offers a more flexible and powerful approach for analyzing complex and nonlinear data patterns compared to linear regression techniques.

  1. It is necessary to describe the dataset directly used for linear regression training in more detail.

Details added. (Table 1.a, 1.b, 1.c)

  1. Panel Data contains categorical values. The article does not clearly describe how categorical data are converted to numerical data and why this particular method is optimal.

Included in 2.1.1. and 2.1.2. parts.

  1. It would be appropriate to expand the overview of the application of the fractal approach in mathematical modeling. For example, pay attention to publications DOI: 10.3390/fractalfract7080584, DOI: 10.1016/j.rinp.2020.103702 and DOI: 10.1007/978-3031-04812-8_9

Included in introduction part.

  1. Conclusion section should be extended using:

- limitations of the proposed approach; - prospects for future research.

The study's limitation revolves around the predictive accuracy of the proposed method beyond a thirty-day forecast for reservations, where deviations greater than 10% may occur. This highlights a potential challenge in achieving reliable long-term forecasting accuracy. To address this limitation, future research could focus on refining the modeling approach to enhance its predictive capability for extended forecasting periods. Possible remedies for improvement may include exploring advanced forecasting techniques, incorporating additional variables or factors into the model, proposing the DAM with first and second derivative and optimizing the methodology to make it more robust and adaptable to longer-term prediction scenarios. By addressing this limitation, future studies could enhance the practical applicability and reliability of the proposed method in real-world air transport capacity management contexts.

Reviewer 2 Report

Comments and Suggestions for Authors

My comments and suggestions are in the attached file. 

Author Response

We would like to express our gratitude to the valuable Reviewer since without his/her remarks, the article would not be complete. Thank you, a lot. 

Reviewer 3 Report

Comments and Suggestions for Authors

The paper thoroughly examines the Passenger Load Factor (PLF) within the aviation industry, proposing a modeling approach employing linear regression and fractional calculus. Here are some comments to be addressed:

-It is suggested that the paper's title and abstract clearly articulate the research question, data sources, and methodology.

-The methods section should elucidate the construction of data frames, calculate load factor lag, and integrate these elements into the analysis.

-Providing detailed information on the specific statistical methods utilized for data analysis, along with clear presentation of results through visualizations, is recommended.

-The discussion section should interpret the findings, expound on their relevance to the research question, and address any study limitations.

-While the paper introduces an innovative approach by incorporating fractional calculus for PLF modeling, it acknowledges that this concept may pose challenges for readers unfamiliar with the topic. Consider offering a concise explanation or referring readers to external sources for clarification to enhance comprehension.

Comments on the Quality of English Language

Minor editing of English language required.

Author Response

Response to Review 3

First, we would like to express our gratitude to the valuable Reviewer since without his/her remarks, the article would not be complete. Thank you, a lot. We will try to answer in order.

The paper thoroughly examines the Passenger Load Factor (PLF) within the aviation industry, proposing a modeling approach employing linear regression and fractional calculus. Here are some comments to be addressed:

-It is suggested that the paper's title and abstract clearly articulate the research question, data sources, and methodology.

Title:

Modeling and Predicting Passenger Load Factor in Air Trans-portation: A Deep Assessment Methodology with Fractional Calculus Approach Utilizing Reservation Data

Abstract:

This study addresses the challenge of predicting Passenger Load Factor (PLF) in air transportation to optimize capacity management and revenue maximization. Leveraging historical reservation data from 19 Turkish Airlines market routes and sample flights, we propose a novel approach combining Deep Assessment Methodology (DAM) with fractional calculus theory. By modeling the relationship between PLF and the number of days remaining until a flight, our method yields minimal errors compared to traditional techniques. Through a continuous curve constructed using the least-squares approach, we enable the anticipation of future flight values. Our analysis demonstrates that the DAM model with a first-order derivative outperforms linear techniques and Fractional Model-3 in both modeling capabilities and prediction accuracy. The proposed approach offers a data-driven solution for efficiently managing air transport capacity, with implications for revenue optimization. Specifically, our modeling findings indicate that the DAM wd model improves prediction accuracy by approximately 0.67 times compared to the DAM model, surpassing fractional model and regression analysis. For the DAM wd Modeling method, the lowest Average Mean Absolute Percentage Error (AMAPE) value achieved is 0.571, showcasing its effectiveness in forecasting flight outcomes.

-The methods section should elucidate the construction of data frames, calculate load factor lag, and integrate these elements into the analysis.

Details added. (Table 1.a, 1.b, 1.c)

-Providing detailed information on the specific statistical methods utilized for data analysis, along with clear presentation of results through visualizations, is recommended.

Detail added to introduction part.

-The discussion section should interpret the findings, expound on their relevance to the research question, and address any study limitations.

The study's limitation revolves around the predictive accuracy of the proposed method beyond a thirty-day forecast for reservations, where deviations greater than 10% may occur. This highlights a potential challenge in achieving reliable long-term forecasting accuracy. To address this limitation, future research could focus on refining the modeling approach to enhance its predictive capability for extended forecasting periods. Possible remedies for improvement may include exploring advanced forecasting techniques, incorporating additional variables or factors into the model, proposing the DAM with first and second derivative and optimizing the methodology to make it more robust and adaptable to longer-term prediction scenarios. By addressing this limitation, future studies could enhance the practical applicability and reliability of the proposed method in real-world air transport capacity management contexts.

-While the paper introduces an innovative approach by incorporating fractional calculus for PLF modeling, it acknowledges that this concept may pose challenges for readers unfamiliar with the topic. Consider offering a concise explanation or referring readers to external sources for clarification to enhance comprehension.

Sun, H., Chang, A., Zhang, Y. and Chen, W. (2019) A review on variable-order fractional differential equations: mathematical foundations, physical models, numerical methods and applications. Fractional Calculus and Applied Analysis, Vol. 22 (Issue 1), pp. 27-59. https://doi.org/10.1515/fca-2019-0003

Podlubny, I. (1998) Fractional Differential Equations: An Introduction to Fractional Derivatives, Fractional Differential Equations, to Methods of Their Solution and Some of Their Applications. Vol. 198, Academic Press, Mathematics in Science and Engineering, 366.

Are added to introduction part and also reference list.
